# Is Cardiopulmonary Exercise Testing Predictive of Surgical Complications in Patients Undergoing Surgery for Ovarian Cancer?

**DOI:** 10.3390/cancers15215185

**Published:** 2023-10-28

**Authors:** Anke Smits, Claire-Marie Agius, Dominic Blake, Christine Ang, Ali Kucukmetin, Maaike van Ham, Johanna M. A. Pijnenborg, Joanne Knight, Stuart Rundle

**Affiliations:** 1Northern Gynaecological Oncology Centre, Queen Elizabeth Hospital, Gateshead NE9 6SX, UK; 2Department of Gynaecological Oncology, Radboudumc, 6525 GA Nijmegen, The Netherlands; 3Department of Anaesthetics, Queen Elizabeth Hospital, Gateshead NE9 6SX, UK

**Keywords:** ovarian cancer, surgery, cardiopulmonary exercise testing, complications

## Abstract

**Simple Summary:**

Treatment of ovarian cancer often involves extensive surgery and is associated with an increased risk of surgical complications. Cardiopulmonary exercise testing (CPET) is a tool used to assess patients’ cardiac and respiratory fitness, and the patient’s ability to withstand the rigor of extensive surgery. Cardiorespiratory fitness is recognised as a predictive tool for surgical complications in many surgical specialities but has not yet been assessed in ovarian cancer patients. The aim of our retrospective study was to evaluate the value of CPET in predicting surgical complications in ovarian cancer surgery. In addition, we assessed the relationship between cardiorespiratory fitness and other outcomes such as hospital stay, readmission and resectability of disease. We found that patients with a raised VE/VCO_2_ observed during CPET, experienced more surgical complications. However, we did not find a relationship with other outcomes. Future studies are needed to further delineate the predictive value of CPET in this population.

**Abstract:**

Preoperative cardiopulmonary exercise testing (CPET) provides an objective assessment of functional capability. In other intra-abdominal surgical specialties, CPET outcomes are predictive of operative morbidity. However, in ovarian cancer surgery, its predictive value remains unknown. In this study, we evaluated the association between CPET performance and surgical morbidity in ovarian cancer patients. Secondly, we assessed the association between CPET performance and other surgical outcomes (i.e., hospital stay, readmission and residual disease). This was a retrospective cohort study of patients undergoing primary surgery for ovarian cancer between 2020 and 2023. CPET performance included peak oxygen uptake (VO_2_ max), ventilatory efficiency (VE/VO_2_) and anaerobic threshold. Outcomes were operative morbidity and included intra- and postoperative complications (Clavien–Dindo), hospital stay, readmission within 30 days and residual disease. A total of 142 patients were included. A lower VO_2_ peak and a higher VE/VCO_2_ were both associated with the occurrence of postoperative complications, and a poorer anaerobic threshold was associated with more transfusions. VE/VCO_2_ remained significantly associated after multivariate analysis (*p* = 0.035). None of the CPET outcomes were associated with length of stay, readmission or residual disease. In conclusion, VE/VCO_2_ was significantly associated with an increased risk of all-cause postoperative complications in ovarian cancer patients undergoing primary surgery.

## 1. Introduction

Sixty percent of patients with ovarian cancer are diagnosed with advanced-stage disease, which is treated through a combination of surgery and systemic therapy. Surgery is often extensive, with the aim of removing all visible disease [1]. Ovarian cancer patients are a high-risk surgical population, characterised by increasing age, poor performance status, high symptom burden, poor nutritional status and a sedentary lifestyle [2,3,4,5]. Consequently, perioperative complications remain prevalent, and pre-operative risk stratification remains challenging in this population [6].

Current practices vary for the pre-operative assessment of patients scheduled for surgery in ovarian cancer. Cardiopulmonary exercise testing (CPET) is used in many centres as an objective measure of functional capacity under stress as part of a comprehensive assessment of the ability of a patient to withstand the rigors of complex extended surgery. It combines an assessment of a patient’s pulmonary and cardiac systems, and includes ECG, lung function tests, blood pressure measurement and continuous saturation assessment along with the measurement of inspired and expired gases during exercise [7,8]. Alterations in CPET outcomes/variables such as anaerobic threshold <10 mL/kg/min, peak VO_2_ < 15 mL/kg/min and VE/VCO_2_ > 34 at the anaerobic threshold are associated with poor postoperative outcomes in elective intra-abdominal surgery [8,9,10]. In addition, recent evidence suggests that CPET may also be of use in predicting specific surgical outcomes such as cytoreductive status in ovarian cancer [11].

At our institution, a high-volume ovarian cancer surgical centre, CPET assessment is used prior to staging and cytoreductive surgery for ovarian cancer. Therefore, in this study, we sought to assess the association between CPET performance and surgical morbidity, hospital stay and readmission in ovarian cancer patients receiving primary surgical treatment. In addition, we assessed the association between CPET performance, other surgical outcomes and patient characteristics.

## 2. Materials and Methods

### 2.1. Study Population

This was a retrospective cohort study of patients undergoing primary surgery for suspected or confirmed ovarian cancer between January 2020 and January 2023 at the Northern Gynaecological Oncology Centre, Gateshead, United Kingdom. Patients with benign or borderline pathology as final histological diagnosis were excluded. In addition, we excluded patients with synchronous tumours outside of the gynaecological tract. All patients who had an anaesthetic assessment with intent of CPET were included. CPET was performed using a standard exercise protocol for perioperative CPET. Most prevalent contra-indications to CPET included myocardial infarction, other severe cardiac pathology and uncontrollable asthma, with relative contra-indications being severe hypertension, recent thrombo-embolic events and arrhythmias [8]. In all cases, primary surgery comprised an exploratory laparotomy and peritoneal assessment. In cases with no peritoneal disease, the mass lesion was used for intra-operative diagnosis through frozen-section analysis to guide surgical management [12]. When frozen-section analysis confirmed malignancy, a standard staging procedure that included contralateral salpingo-oophorectomy and total hysterectomy, infra-colic omentectomy, bilateral systematic pelvic lymphadenectomy +/− infra-renal para-aortic lymphadenectomy was performed. In cases where peritoneal disease was identified intra-operatively or pre-operatively, maximum-effort cytoreduction was performed. Intra-operative diagnosis was utilised if the result would alter surgical management. This was a secondary analysis of an established continuous audit of practice and service evaluation as a referral centre for ovarian cancer, and, as such, was exempt from ethical approval.

### 2.2. Data Collection

Patient data were collected retrospectively. Demographic and clinical characteristics of patients were collected from patient’s medical records. Baseline characteristics included age at diagnosis, previous medical history (including cardiovascular disease, diabetes, pulmonary disease and other comorbidities) stratified according to the Charlson Comorbidity Index (CCI), Eastern Cooperative Oncology Group (ECOG) performance status, Rockwood frailty index, nutritional status, body mass index (BMI) and smoking status [13,14,15]. Clinical characteristics included histological subtype, grade and clinical FIGO stage (2014), and type of surgery.

CPET was performed and interpreted by consultant anaesthetists trained through the Perioperative Exercise Testing and Training Society (POETTS). The standard operating procedure for conducting the test followed the POETTS guidelines [8]. Exercise testing was conducted on an electromagnetically braked cycle ergometer (Ergoselect 200). Ventilation and gas exchange were measured using the UltimaTM CardiO_2_ System metabolic cart. Resting spirometry was routinely performed except between March 2020 and July 2022 due to concerns regarding aerosol-generating procedures during the SARS-CoV-2 pandemic. CPET data were analysed using the Breeze 7.2.0.64 SP7 and CardioControlWorkstation Software. CPET outcomes included anaerobic threshold (AT, mL/min^1^), peak oxygen uptake (VO_2_, mL/kg/min) and ventilatory efficiency for carbon dioxide (VE/VCO_2_) at anaerobic threshold and were risk-stratified (AT < 10 mL/min^1^, VO_2_ < 15 mL/kg/min and VE/VCO_2_ > 34 being at higher risk) [8]. CPET outcomes were classified as low-, intermediate- and high-risk as per local guidelines [16]. The American Society of Anaesthesiologists (ASA) physical status classification was assigned by the anaesthesiologist during the pre-operative screening.

### 2.3. Outcomes

Primary outcomes included operative morbidity and mortality, length of hospital stay and readmission within 30 days. Operative morbidity included intra-operative complications such as total blood loss, red blood cell transfusion, injuries to vessels, nerves and urinary tract, and postoperative complications such as fever, infection, urinary tract infection, wound problems, ileus, thrombo-embolisms, transfusion requirements and re-interventions. Morbidity was graded according to the Clavien–Dindo classification system [17]. Secondary outcomes were residual disease (cytoreductive status: no residual disease (R0), macroscopic residual disease with a diameter of 0.1–1 cm (R1) or >1 cm (R2)), surgical complexity score in ovarian cancer surgery, operating time and estimated blood loss [18].

### 2.4. Statistical Analyses

Continuous variables are presented as means with standard deviations or medians and interquartile range, as appropriate. Categorical outcomes are presented as frequencies and proportions. Continuous data were analysed using non-parametric tests (Mann–Whitney U test), with pairwise comparisons where appropriate, and categorical data were analysed using Pearson’s chi-squared tests and Fisher’s exact tests. Logistic regression models were used while controlling for possible confounders for binary outcome data. Statistical tests were two-tailed and considered significant at *p* < 0.05. Data were analysed using IBM SPSS Statistics for Windows, version 25 (IBM Corp., Armonk, NY, USA).

## 3. Results

A total of 181 patients underwent primary surgery for ovarian cancer between January 2020 and January 2023 at the Northern Gynaecological Oncology Centre. Of these patients, 175 patients were eligible, excluding patients whose files were unavailable (N = 6). Thirty additional patients were excluded as no CPET was performed. Reasons for not performing CPET are detailed in Figure 1. The final study population included 142 patients.

Baseline and clinical characteristics of the study population are detailed in Table 1. The median age of patients was 62 years, 39.4% were overweight and another 27.4% were obese. The majority had an ECOG performance score of 0 (64.8%). Almost half of the patients had a low CCI (43.0%) and just under one-fifth were found to have high or very high CCI (17.2%). Sixty-seven patients (47%) had a Rockwood frailty score of 1 to 3, 23 patients (16.2%) had a score of 4 or higher, and in 52 patients (36.6%), frailty was not documented. The majority of patients were diagnosed with advanced-stage disease (III–IV, 64.8%) and high-grade serous adenocarcinoma was the most prevalent histological subtype.

Of patients who attempted CPET, 136/142 (96%) successfully completed the test. Six patients did not complete the test through failure to reach the anaerobic threshold due to exhaustion (N = 3), joint pain (N = 1) or unknown reasons (N = 2). CPET outcomes are detailed in Table 2. Forty-six (32.4%) patients had a peak VO_2_ uptake of less than 15 mL/kg/min, 24 patients (16.9%) had a VE/VCO_2_ of >34 and 32 patients (22.5%) had an anaerobic threshold under 10 mL/min^1^. Less than half of the patients (49.3%) were categorised as low-risk for perioperative morbidity and mortality.

CPET outcomes including VO_2_ peak, VE/VCO_2_ and anaerobic threshold were associated with poor patient clinical characteristics (Table 3). VO_2_ peak < 15 mL/kg/min was significantly associated with a high or very high index of morbidity based on pre-operative CCI score (*p* < 0.009), a higher BMI (*p* < 0001) and a poorer ASA status (*p* = 0.001). The VO_2_ peak as a continuous variable showed an inverse association with frailty (*p* = 0.004). Rockwood frailty scores of 4 or above (vulnerable or worse) also predicted poorer performance during CPET using VE/VCO_2_ (*p* < 0.001) and AT (*p* < 0.001) as continuous variables. Measures of good pre-morbid functioning predicted better test performance by a lower VE/VCO_2_ (<35): these included a better ECOG performance status (*p* < 0.001) or ASA score (*p* = 0.022), and less severe frailty (*p* = 0.010). Conversely, increasing age was associated with a higher VE/VCO_2_ (>35) and therefore a worse test performance. Higher BMI was also associated with a higher VE/VCO_2_ as a continuous variable (*p* = 0.022). An anaerobic threshold of <10 mL/min^1^ was associated with a higher BMI (BMI > 30 kg/m^2^, *p* < 0.001) and a higher ASA status (*p* = 0.013). In addition, a Rockwood score of 4 or more was associated with a lower anaerobic threshold (*p* = 0.004).

Sixty-three patients (44.4%) underwent a laparotomy with a frozen section of an ovarian mass for suspected ovarian cancer, of which seventeen patients (12.0%) had an intra-operative finding of unexpected disseminated disease (Table 4). Furthermore, 79 patients (56.6%) underwent a cytoreductive surgery for ovarian cancer. Surgical complexity scores for ovarian cancer surgery are detailed in Table 4. In 85.9% of patients, complete cytoreduction (no residual disease) was achieved. Fifteen (10.6%) patients had an intra-operative complication and 30 patients (21.1%) received a blood transfusion. Nighty-nine (69.7%) patients had a postoperative complication, of which the majority (92.9%) were Clavien–Dindo 1–2 complications. The most prevalent complications were infectious complications (N = 73), of which 27 patients had a surgical site infection (SSI) and 17 patients had a fever without a source. Twenty-nine patients received a blood transfusion postoperatively. One patient died in the postoperative period due to pulmonary embolism leading to cardiac arrest.

### CPET Outcomes, Surgical Morbidity and Other Outcomes

Poor CPET test performance assessed according to a lower VO_2_ peak and a higher VE/VCO_2_ was both associated with the occurrence of postoperative complications (*p* = 0.038 and *p* = 0.004, respectively). In addition, a higher VE/VCO_2_ was specifically associated with infectious complications (*p* = 0.039). Multivariate analyses showed that only VE/VCO_2_ remained significantly associated (*p* = 0.035) with postoperative complications after correcting for other characteristics (age, BMI, frailty, ASA status and VO_2_ peak). Previously defined cut-off values of VO_2_ peak <15 mL/kg/min and VE/VCO_2_ > 34 did not show significant associations with surgical complications or postoperative morbidity in this patient population. An anaerobic threshold of <10 mL/min^1^ was significantly associated with postoperative transfusion rates, with 34.4% transfusion rates in patients with a threshold of <10 mL/min^1^ versus 17.3% in the >10 mL/min^1^ group (*p* = 0.050) in univariate analysis, but not after multivariate analysis. The CPET outcome parameters did not show any other associations with intra-operative complications, transfusion rates, readmission or hospital stay. Subgroup analyses of early (I–II) and advanced (III–IV) stage disease and those with a higher surgical complexity score (>8) revealed no additional associations between CPET outcomes and operative morbidity, hospital stay or readmissions.

With respect to the secondary outcomes, the cytoreduction status, i.e., residual disease, was not associated with any of the CPET outcomes (VO_2_ peak *p* = 0.507, VE/VCO_2_
*p* = 0.549, AT *p* = 0.420), nor were surgical complexity or duration of surgery associated.

## 4. Discussion

We demonstrated in our study cohort that VE/VCO_2_ was the only objective CPET outcome measurement associated with an increased risk of all-cause postoperative morbidity in patients undergoing laparotomy for ovarian cancer. The VO_2_ peak and anaerobic threshold did not show a significant association with intra- and postoperative complications. Furthermore, CPET outcomes were not associated with length of hospital stay, readmission, residual disease or surgical complexity.

The rationale behind our current practices relating to CPET testing of ovarian cancer surgery candidates lies in the idea that surgical stress increases the oxygen demands of tissue and organs, with patients with poor exercise capacity struggling to meet this. Consequently, patients with a demonstrated reduction in the ability to meet the physiological demands of exercise should be at higher risk for adverse perioperative outcomes. Poorer capacity may be the result from deconditioning due to the disease process and disease burden, but also due to advancing age and comorbidities [19]. In this study, we showed that VE/VO_2_ is associated with perioperative morbidity but failed to show further associations between CPET performance, operative complications and hospital stay.

Whilst, historically, measures of oxygen uptake have been seen as the primary outcome in assessing fitness for abdominal surgery, emerging evidence from other surgical specialities implicates ventilatory inefficiency, VE/VCO_2_, as a significant predictor of postoperative outcomes. A raised VE/VCO_2_ but not reduced AT or peak VO_2_ was found to be a predictive marker of postoperative mortality in patients undergoing hepatobiliary surgery [20]. In radical cystectomies, VE/VCO_2_ > 33 was the most significant determinant of postoperative complications and increased length of stay [21]. In colorectal cancer surgery, VE/VCO_2_ > 39 predicted death at 90 days [22].

In ovarian cancer, there is a complex interrelationship between the physiological burden of often-extensive disease, its compound effects on nutritional status and poor physical fitness, which may influence CPET performance in the pre-treatment setting [2,5,23]. This additional complexity and the heterogeneity in the patient population are likely to underlie the relative lack of published data regarding CPET outcomes and associations with perioperative morbidity in ovarian cancer. Only one study has evaluated CPET performance and surgical morbidity in ovarian cancer. Element et al. assessed 43 patients undergoing cytoreductive surgery (primary N = 17, interval N = 26) for advanced ovarian cancer, and concurringly reported no associations between CPET outcomes (anaerobic threshold <11 and VO_2_ max < 15 mL/kg/min) and operative complications or readmissions. However, they did report that an aerobic threshold of ≥11 mL/min^1^ was associated with higher rates of R0 and R1 cytoreduction, and higher surgical complexity scores. In addition, patients with a VO_2_ max of ≥15 mL/kg/min also received surgery with a higher complexity score. They concluded that patients with poor CPET performance are more likely to receive suboptimal cytoreductive outcomes from surgery [11]. However, there were significant differences in age and comorbidities between low- and high-risk CPET groups, which was not corrected through multivariate analyses. In addition, the authors themselves postulated that poorer CPET outcomes may have influenced surgical decision making with respect to the extent of radical surgery undertaken.

The hypothesis that operative morbidity is higher among unfit patients is intuitively appealing. Evidence from studies in patients undergoing surgery for other intra-abdominal pathology supports the use of CPET as a predictive measure of operative morbidity complementary to its primary assessment of fitness [8,10,24]. Moran et al. assessed the ability of CPET to predict postoperative outcomes from 37 studies (N = 7852), including general intra-abdominal surgery, hepatic, colorectal, gastro-intestinal, pancreatic, bariatric and renal surgery. They reported that a reduced anaerobic threshold was associated with 30-day mortality and that all three CPET parameters were to varying degrees associated with postoperative morbidity. In addition, a poorer anaerobic threshold was also associated with a prolonged hospital stay in most studies. In the four studies in patients undergoing intra-abdominal surgery, anaerobic threshold was the best overall predictor of morbidity [10]. The discrepancy between these findings and our findings may be due to study size but may also be explained by a difference in populations as the majority included non-cancer surgery [10]. In addition, previous systematic reviews have shown that the optimal CPET predictors of high risk appears to vary depending on the type, extent and indication for intended surgery [10,24].

There is a recognition that pre-operative objective surgical risk stratification is desirable for the treatment of ovarian cancer. Complete macroscopic resection of a disease is the most important prognostic factor for the patient. Surgery is often consequently extensive with a relatively high operative morbidity in comparison to other types of surgery [1,25]. Patient selection for the most radical of procedures is therefore crucial to determining optimal primary treatment strategies. Many patient factors have been identified as contributors to surgical risk, such as age, BMI, albumin levels and performance status [4,25]. More objective measures of functional capacity, such as the six-minute walk test or stair climbing assessment, have been considered as aides to surgical decision making in cancer treatments, but there is a scarcity of data when considering their use in the ovarian cancer population [26,27].

Our results do not indicate that CPET was used as a tool by which to refine the intended radicality of staging or cytoreductive surgery, but was mainly used as an adjunct to assess perioperative risk. CPET can contribute to the decision-making process when exploring treatment options for ovarian cancer, as well as direct perioperative monitoring and postoperative care requirements. Poor CPET outcomes were not an absolute contraindication to surgery. Once the decision for surgical treatment was reached, the radicality of surgery was defined by the procedures required for complete staging or macroscopic complete resection of disease and the CPET parameters used to guide intra-operative and immediate postoperative care in the context of the surgery performed.

Our data also show that CPET performance was associated with several patient characteristics such as comorbidities, BMI and frailty. This is consistent with previously published evidence that suggests frailty as an independent predictor of complications in highly complex ovarian cancer surgery [28]. Future research should therefore further evaluate the relationships between these adverse pre-morbid patient factors and operative outcomes with the aim of identifying threshold variables specific for risk stratification in ovarian cancer patients. This strategy is particularly attractive in the context of the resource-limited health system due to the cost implications of universal CPET testing. In addition, CPET remains a strenuous undertaking for patients, burdening them with fatigue, exhaustion and breathing issues on top of their symptoms [29]. Prehabilitation programmes prior to ovarian cancer treatment are also gaining traction internationally [30]. Further research may help identify modifiable patient characteristics for prehabilitation interventions to reduce the functional decline caused by the burden of cancer and its treatments.

This is the largest study to date assessing the associations between CPET outcomes and morbidity in ovarian cancer patients undergoing primary surgery. Strengths of this study are the correction for patient factors known to be associated with morbidity, and that we attempted to reduce the heterogeneity of the study population by only including patients undergoing primary surgery, or who were treatment-naïve. However, this study is limited by its retrospective design, which includes a possible selection bias and the completeness of previously recorded data. Future studies with prospective designs are required to further evaluate these associations. In addition, we included all-stage ovarian cancer patients, although secondary analyses showed no further association when separately assessing early- and advanced-stage disease.

This study failed to show an association between CPET outcomes and either length of hospital stay or readmission. The discrepancy between these findings and those of higher complication rates in patients with poor ventilatory parameters may be explained by the high proportion of low-grade infectious morbidity in this population, as these types of complications are less likely to have resulted in an additional length of stay. In addition, a low threshold for recording, the early treatment of suspected SSI in the absence of microbiological confirmation and the thorough collection of data of Clavien–Dindo grade 1 complications are also likely to have contributed to these findings. We note, however, that the reported serious postoperative complication rate of 5% (Clavien–Dindo 3–5) does concur with the current literature [6,31].

## 5. Conclusions

We observed that an increased VE/VCO_2_ is associated with an increased risk of postoperative complications in patients with ovarian cancer undergoing primary surgery. However, we did not find an association between other CPET outcomes and surgical morbidity and hospital stay. Importantly, poorer CPET performance did not prohibit maximal-effort cytoreductive surgery. Future studies need to further assess the predictive value of CPET performance on surgical morbidity and other outcomes whilst evaluating clinical stratification as an alternative for use in resource-limited settings. The current body of evidence is insufficient to draw any definitive decisions on the value of CPET beyond part of the anaesthetic assessment.

## Figures and Tables

**Figure 1 cancers-15-05185-f001:**
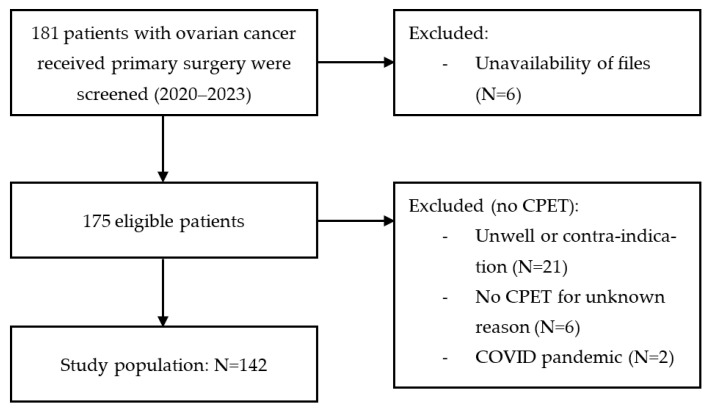
Patient selection process.

**Table 1 cancers-15-05185-t001:** Baseline and clinical characteristics of the study population.

Characteristics	Study Population N 142 (%)
Age in years (median, range)	62 (30–85)
Ethnicity	
British	30 (21.1%)
White British	14 (9.9%)
Other	1 (0.7%)
Unknown	97 (68.3%)
ECOG	
0	92 (64.8%)
1	39 (27.5%)
2	8 (5.6%)
Unknown	3 (2.1%)
BMI (kg/m^2^)	
Underweight (<18.5)	2 (1.4%)
Normal (18.5–24.9)	45 (31.7%)
Overweight (25–29.9)	56 (39.4%)
Obese (30–39.9)	34 (23.9%)
Morbidly obese (≥40)	5 (3.5%)
Charlson Comorbidity Index	
Low (0)	61 (43.0%)
Medium (1–2)	56 (39.4%)
High (3–4)	19 (13.4%)
Very high (≥5)	6 (4.2%)
Smoking	
Yes	16 (11.3%)
No	125 (88.0%)
Unknown	1 (0.7%)
ASA score	
1	6 (4.2%)
2	88 (62.0%)
3	48 (33.8%)
Frailty score	
Rockwood 1	4 (2.8%)
Rockwood 2	30 (21.1%)
Rockwood 3	33 (23.2%)
Rockwood 4	17 (12.0%)
Rockwood 5	4 (2.8%)
Rockwood 6	2 (1.4%)
Unknown	52 (36.6%)
Stage	
I	34 (23.9%)
II	16 (11.3%)
III	70 (49.3%)
IIIa	12 (8.5%)
IIIb	10 (7.0%)
IIIc	48 (33.8%)
IV	22 (15.5%)
IVa	2 (1.4%)
IVb	20 (14.1%)
Histology	
High grade serous	77 (55.6%)
Mucinous	17 (12.0%)
Clear cell	12 (8.5%)
Low grade serous	13 (9.2%)
Endometrioid	7 (4.9%)
Carcinosarcoma	2 (1.4%)
Other	12 (8.5%)

**Table 2 cancers-15-05185-t002:** CPET outcomes of the population.

CPET Outcomes	Study Population N (%)
VO_2_ Peak (mL/kg/min)	
<15	46 (32.4%)
≥15	95 (67.4%)
Unknown	1 (0.7%)
VE/VCO_2_	
≤34	116 (81.7%)
>34	24 (16.9%)
Unknown	2 (1.4%)
Anaerobic threshold (mL/min^1^)	
<10	32 (22.5%)
≥10	104 (73.2%)
Not reached/unknown	6 (4.2%)
Risk category	
Low	70 (49.3%)
Intermediate	29 (20.4%)
High	38 (26.8%)
Unknown	5 (3.5%%)

**Table 3 cancers-15-05185-t003:** Association between baseline characteristics and CPET.

	VO_2_ Peak	VO_2_ Peak	VE/VCO_2_	VE/VCO_2_	AT	AT
	<15 vs. ≥15 (mL/kg/min)	(Continuous, (mL/kg/min)	<35 vs. ≥35	(Continuous)	<10 vs. ≥10 (mL/min^1^)	Continuous (mL/min^1^)
Age	0.148	Not performed	<0.001 *	Not performed	0.857	Not performed
ECOG performance (0–2)	0.782	0.308	0.295	<0.001 *	0.305	0.662
CCI Low and medium vs. high and very high	0.009 * Low and medium: 27.6% < 15 High and very high: 56.0% < 15	0.001 *	0.770	0.066	0.287	0.182
BMI <30 vs. ≥30	<0.001 *	<0.001 *	0.313	0.022 *	<0.001 *	<0.001 *
	<30: 22.5% < 15				<30: 15.8% < 10	
	≥30: 59.0% < 15				≥30: 45.7% < 10	
Frailty Rockwood 1–3 vs. ≥4	0.138	0.004 *	0.010 *	<0.001 *	0.067	0.006 *
			1–3: 87.9% < 35		1–3: 18.2% < 10	
			≥4: 59.1% < 35		≥4: 40.0% < 10	
ASA status (1–3)	0.001 *	<0.001 *	0.022 *	0.027 *	0.012 *	0.003 *
	1: 0% < 15		1: 100% < 35		1: 0% < 10	
	2: 24.1% < 15		2: 88.5% < 35		2: 17.4% < 10	
	3: 52.1% < 15		3: 70.2% < 35		3: 38.6% < 10	

AT: anaerobic threshold; ASA: the American Society of Anesthesiologists status. BMI: body mass index; CCI: Charlson Comorbidity Index; ECOG: Eastern Cooperative Oncology Group; *: *p* < 0.05

**Table 4 cancers-15-05185-t004:** Surgical details, outcomes and complications of the population.

Characteristics	Study Population N 142 (%)
Surgery	
Laparotomy with frozen section	63 (44.4%)
Cytoreductive surgery	79 (55.6%)
Surgical complexity scores	
Low (0–3)	44 (31.0%)
Intermediate (4–7)	77 (54.2%)
High (>7)	21 (14.8%)
Operating time (mins, median, range)	275 (95–526)
Estimated blood loss (mL, median, range)	800 (10–6500)
Cytoreduction	
No residual disease (R0)	122 (85.9%)
Residual disease 0.1 to 1 cm (R1)	12 (8.5%)
Residual disease of >1 cm (R2)	8 (5.6%)
Intra-operative complications	
Yes	15 (10.6%)
Bladder injury	5 (3.5%)
Ureteric injury	4 (2.8%)
Vessel injury	(2.8%)
Splenic tear	1 (0.7%)
Pneumothorax	1 (0.7%)
No	127 (89.4%)
Intra-operative transfusion	
Yes	30 (21.1%)
No	112 (78.9%)
Postoperative complications	
Yes	99 (69.7%)
No	43 (30.3%)
Postoperative complications (CDC)	
1	24 (16.9%)
2	68 (47.9%)
3	6 (4.2%)
4	0 (0%)
5	1 (0.7%)
None	43 (30.3%)
Infectious complications	
Yes	73 (51.4%)
Surgical site infection	27 (19.0%)
Fever e.c.i.	17 (12.0%)
Infection elsewhere	29 (20.4%)
No	69 (48.6%)
Postoperative transfusion	
Yes	29 (20.4%)
No	113 (79.6%)
Postoperative ward	
HDU	91 (64.1%)
Ward	51 (35.9%)
Hospital stay (mean, range)	8 (1–71)
Readmission within 30 days	
Yes	11 (7.7%)
No	131 (92.3%)

CDC: Clavien–Dindo classification; e.c.i.: causa ignota.

## Data Availability

The data presented in this study are available in this article.

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
