# Peer review of "Is Cardiopulmonary Exercise Testing Predictive of Surgical Complications in Patients Undergoing Surgery for Ovarian Cancer?"

_cancers, 2023, doi:10.3390/cancers15215185_

Round 1

Reviewer 1 Report

Comments and Suggestions for Authors

The present retrospective study by Smits A et al evaluates the association between cardiopulmonary exercise testing (CPET) performance and surgical morbidity in ovarian cancer patients. It is well established that functional capacity is a very strong predictor of the outcome in various disease settings. Moreover, CPET is generally accepted as the gold standard for functional capacity assessment. In general, the present study adds to the aforementioned notion in ovarian cancer patients undergoing laparotomy. Overall the study is concise and well-written. Yet, according to the results none of the CPET outcomes were associated with length of stay or readmission. I think that the authors should comment more on this phenomenically unexpected finding.  

Author Response

The present retrospective study by Smits A et al evaluates the association between cardiopulmonary exercise testing (CPET) performance and surgical morbidity in ovarian cancer patients. It is well established that functional capacity is a very strong predictor of the outcome in various disease settings. Moreover, CPET is generally accepted as the gold standard for functional capacity assessment. In general, the present study adds to the aforementioned notion in ovarian cancer patients undergoing laparotomy. Overall the study is concise and well-written. Yet, according to the results none of the CPET outcomes were associated with length of stay or readmission. I think that the authors should comment more on this phenomenically unexpected finding. 

Response – We thank the reviewer for their generous appraisal of our manuscript. We agree that the finding of no association between CPET outcomes and either length of stay or readmission to hospital are surprising and more so due to the apparent association between ventilatory inefficiency, as measured by VE/VCO2, and all-class post-operative complications. It is difficult to comment extensively on the absence of a finding (no association between CPET outcomes and LoS/readmission) as is suggested in the comment, particularly when the basis for the formulation of the research question is that there is a lack of existing evidence in ovarian cancer that suggests CPET should be a gold standard test.

We have, however discussed the potential weaknesses of the study including its retrospective design, the heterogeneity of stage, surgical radicality and patient demographics (see lines 326-330 in original manuscript). We have also modified the last paragraph of the discussion section to address the reviewers comments: stating the surprising absence of the expected finding and attempting to explain (within the limitations of the data presented) the apparent contradiction that high complication rates in those with poor ventilatory outcomes do not translate into longer stay or readmission (Highlighted section 331-338 on revised manuscript).

This contradiction is likely due to a preponderance of low grade (Grade 1 and Grade 2) complications (particularly surgical site infection/infectious complications) that did not require additional inpatient hospital stay.

Reviewer 2 Report

Comments and Suggestions for Authors

This is a retrospective cohort study of cardiopulmonary exercise testing for the preoperative evaluation of ovarian cancer patients that will undergo primary debulking surgery. The methodology is sound, english language and grammar is without problems. 

Author Response

This is a retrospective cohort study of cardiopulmonary exercise testing for the preoperative evaluation of ovarian cancer patients that will undergo primary debulking surgery. The methodology is sound, english language and grammar is without problems.

Response – We thank the reviewer for their kind remarks and generous evaluation of our manuscript. We hope that they find the revised manuscript, taking into account modifications recommended by colleague reviewers to be equally satisfactory.

Reviewer 3 Report

Comments and Suggestions for Authors

I read with great interest the Manuscript titled " Is cardiopulmonary exercise testing predictive of surgical complications in patients undergoing surgery for ovarian cancer? " which falls within the aim of the Journal.

Although the manuscript can be considered already of good quality, I would suggest following recommendations:

-       I suggest a round of language revision, in order to correct few typos and improve readability.

-       The authors should discuss topic and results of this study considering evaluation of various comorbidity and older age for each patient, and the possible application of the frailty index to predict postoperative morbidity and mortality. I suggest authors to read and insert in references the following article PMID: 33223220

-       Considering topic and result of this study, authors could extended the discussion by evaluating the role of lymphadenectomy in advanced ovarian cancer, highlighting the advantages and limitations of this procedure. For example, I suggest this article to get deeper in the topic PMID:  32036457)

Because of these reasons, the article should be revised and completed. Considering all these points, I think it could be of interest to the readers and, in my opinion, it deserves the priority to be published after minor revisions.

Comments on the Quality of English Language

 I suggest a round of language revision, in order to correct few typos and improve readability.

Author Response

I read with great interest the Manuscript titled " Is cardiopulmonary exercise testing predictive of surgical complications in patients undergoing surgery for ovarian cancer? " which falls within the aim of the Journal.

Although the manuscript can be considered already of good quality, I would suggest following recommendations:

Overall response – We thank the reviewer for the care that they have taken in reviewing the manuscript and the advice that they have offered to enhance the final version. We have included specific responses to the reviewer’s comments, below.

-       I suggest a round of language revision, in order to correct few typos and improve readability.

Specific response – We have reviewed the manuscript and corrected any identified typographical errors (highlighted). The manuscript has been reviewed by all authors, four of whom are native English speakers and experienced scientific writers. In the absence of specific detail relating to poor readability we are unable to address this further.

-       The authors should discuss topic and results of this study considering evaluation of various comorbidity and older age for each patient, and the possible application of the frailty index to predict postoperative morbidity and mortality. I suggest authors to read and insert in references the following article PMID: 33223220

Specific response  - Thank you for the direction and supplied reference., We have added a comment into the discussion of findings and referenced the paper at the appropriate point (highlighted, lines 309-311 on revised manuscript)

-       Considering topic and result of this study, authors could extended the discussion by evaluating the role of lymphadenectomy in advanced ovarian cancer, highlighting the advantages and limitations of this procedure. For example, I suggest this article to get deeper in the topic PMID:  32036457)

Specific response – We have reviewed the supplied reference in the context of our study in which the research question is clearly outlined:

 ‘in this study we sought to assess the association between CPET performance and surgical morbidity, hospital stay and readmission in ovarian cancer patients receiving primary surgical treatment.’

The purpose of this study was not to assess the role of lymphadenectomy in the treatment of advanced stage ovarian cancer. The undertaking of lymphadenectomy for complete staging of apparent early stage ovarian cancer is outlined in the methods section (lines 82 – 85) and we have established also that advanced stage disease was treated with maximal effort surgery. We have no data that would be valid in creating groups of lymphadenectomy vs no lymphadenectomy in otherwise comparable patients and therefore find that the discussion of this would not be in keeping with, or supported by any data that we could present. We have therefore respectfully declined to create the discussion that would be necessary to include the reference.

Because of these reasons, the article should be revised and completed. Considering all these points, I think it could be of interest to the readers and, in my opinion, it deserves the priority to be published after minor revisions

Specific response – We hope the reviewer and editors find the revisions adequate and the justification for non-inclusion of the second suggested reference acceptable.

Reviewer 4 Report

Comments and Suggestions for Authors

The authors analyze the value of Cardiopulmonary exercise testing (CPET) as a predictive tool of minor and major complications in patients with ovarian cancer undergoing primary surgery. To our knowledge it is the largest study on this topic; however there are several biases:

- the retrospective nature is a limit; validation of predictive tests of surgical complications should be based on prospective designs

- the results of the study are limited and not very reproducible; only VE/VCO2 is associated with increased risk of postoperative complications; there is no association between other CPET outcomes and surgical morbidity and hospital stay

-  the authors include all stages; although the subgroup analysis does not show differences by stage, it is clear that it is a oversight to include all stages because ovarian cancer represents an extremely heterogeneous disease with burden differences between stages

- 15.5% of stages are IV FIGO. Which type of procedures were performed on these patients? The differences in surgical procedures between patients is an remarkable bias in predictive value of a test

- 44.4% of patients underwent laparotomy with frozen section. Are they a diagnostic laparotomy? How can it be included in patients undergoing highly complex primary cytoreductive surgery? Were no diagnostic laparoscopies performed?

- the relationship between disease burden, ascites, extent of peritoneal involvement, nutritional status is extremely heterogeneous in ovarian cancer to determine a useful and reproducible predictive cardiorespiratory tool

Author Response

The authors analyze the value of Cardiopulmonary exercise testing (CPET) as a predictive tool of minor and major complications in patients with ovarian cancer undergoing primary surgery. To our knowledge it is the largest study on this topic; however there are several biases:

Overall response – We thank the reviewer for the time and care taken to review our manuscript. Please find specific responses to the comments raised, below.

- the retrospective nature is a limit; validation of predictive tests of surgical complications should be based on prospective designs

Specific response – we agree with the reviewer that ideally studies should be based on prospective designs. We have therefore been explicit in methodology and documented our appraisal of the retrospective nature of the study as a weakness. We have added a line specifically stating the need for prospective study designs (line 328-329).

- the results of the study are limited and not very reproducible; only VE/VCO2 is associated with increased risk of postoperative complications; there is no association between other CPET outcomes and surgical morbidity and hospital stay

Specific response – We respectfully disagree with the reviewer’s comment regarding reproducibility. Whilst this is a retrospective study, it is derived from patient data. Other centres with similar practice and data collection could reproduce the analysis (we could not know if the results are reproducible, such is the nature of research) using the extensive methodological description supplied within the manuscript.

-  the authors include all stages; although the subgroup analysis does not show differences by stage, it is clear that it is a oversight to include all stages because ovarian cancer represents an extremely heterogeneous disease with burden differences between stages

Specific response – we agree with the reviewer that there is a difference in surgery among stage groups of ovarian cancer. We have therefore explicitly detailed the potential impact of the heterogeneity of the patient group (particularly in relation to stage) and we have attempted to control for this by stratifying the analysis by stage, also taking into account surgical complexity, and have mentioned it as a weakness in the discussion.

- 15.5% of stages are IV FIGO. Which type of procedures were performed on these patients? The differences in surgical procedures between patients is an remarkable bias in predictive value of a test

Specific response – As detailed in the methodology section (line 86), maximum effort cytoreductive surgery (with the intention of complete macroscopic resection of abdomino-pelvic peritoneal disease) was undertaken for all cases of advanced ovarian cancer included in the study. Stage 4 disease is not a contraindication to primary cytoreductive surgery and is routine practice in our institution provided that the pre-operative evaluation of disease does not suggest sites that are unresectable, as outlined by international (ESGO) guidance and quality indicator publications.

As previously outlined (see prior response) we have been candid about the potential impact of heterogeneity of stage/disease extent, however we must also consider that across stages there may also be heterogeneity of surgical radicality. We have attempted to control for this by analysis stratified by stage and surgical complexity but it did not change the outcomes (lines 221 – 225). To narrow the study to include only high stage or high complexity surgery would limit the validity of the results in a no less impactful way.

- 44.4% of patients underwent laparotomy with frozen section. Are they a diagnostic laparotomy? How can it be included in patients undergoing highly complex primary cytoreductive surgery? Were no diagnostic laparoscopies performed?

Specific response – we agree the reviewer that there is a certain heterogeneity due to the inclusion of all stages. A laparotomy with frozen section is performed in patients who are suspected to have a possible malignant pelvic mass, but no radiological evidence of spread or a proven histological diagnosis prior to surgery. It is not meant as a diagnostic laparotomy, as it also includes a staging procedure as detailed in the methods, or a possible debulking surgery if there is intra-operative evidence of abdominal spread. As even a staging procedure still encompasses a large midline incision and complex surgery (with pelvic and para-aortic lymphadenectomy, possible extensive adhesiolysis and removal of large masses), we have decided to include these stages in the population to evaluate the association between CPET outcomes and operative outcomes. Recognising possible disparities between stage groups, we therefore have performed a subanalysis of groups, but this did not reveal different outcomes.

Diagnostic laparoscopy is not routinely practiced in our institution for the evaluation of pelvic masses nor if there is no radiological evidence of irresectable disease in advanced stage disease.

- the relationship between disease burden, ascites, extent of peritoneal involvement, nutritional status is extremely heterogeneous in ovarian cancer to determine a useful and reproducible predictive cardiorespiratory too.

Specific response – we agree with the difficulty of  preoperative risk stratification of ovarian cancer patients due to the heterogeneity in this group of patients. This is why we have evaluated CPET as an objective measure to predict postoperative outcomes. More, and ideally, prospective studies are needed to evaluate this further, and we hope that our study will inspire future projects.  

Round 2

Reviewer 1 Report

Comments and Suggestions for Authors

The authors have satisfactorily addressed most of my concerns and in particular, the lack of  association between CPET outcomes and either length of stay or readmission. In my opinion, the manuscript has been improved substantially and is acceptable for publication in Cancers.

Reviewer 4 Report

Comments and Suggestions for Authors

Thanks for the quick review and addressing my concerns.
In my opinion, my lack of support for acceptance stems from this study has biases that cannot be overcome and also it has limited scientific interest.